# POLD1 as a Prognostic Biomarker Correlated with Cell Proliferation and Immune Infiltration in Clear Cell Renal Cell Carcinoma

**DOI:** 10.3390/ijms24076849

**Published:** 2023-04-06

**Authors:** Junjie Tian, Cheng Cheng, Jianguo Gao, Guanghou Fu, Zhijie Xu, Xiaoyi Chen, Yunfei Wu, Baiye Jin

**Affiliations:** 1Department of Urology, The First Affiliated Hospital, School of Medicine, Zhejiang University, Hangzhou 310024, China; 2Zhejiang Engineering Research Center for Urinary Bladder Carcinoma Innovation Diagnosis and Treatment, Hangzhou 310024, China

**Keywords:** clear cell renal cell carcinoma, POLD1, cell proliferation, immune infiltration, prognosis

## Abstract

DNA polymerase delta 1 catalytic subunit (POLD1) plays a vital role in genomic copy with high fidelity and DNA damage repair processes. However, the prognostic value of POLD1 and its relationship with tumor immunity in clear cell renal cell carcinoma (ccRCC) remains to be further explored. Transcriptional data sets and clinical information were obtained from the TCGA, ICGC, and GEO databases. Differentially expressed genes (DEGs) were derived from the comparison between the low and high POLD1 expression groups in the TCGA–KIRC cohort. KEGG and gene ontology (GO) analyses were performed for those DEGs to explore the potential influence of POLD1 on the biological behaviors of ccRCC. The prognostic clinical value and mutational characteristics of patients were described and analyzed according to the POLD1 expression levels. TIMER and TISIDB databases were utilized to comprehensively investigate the potential relevance between the POLD1 levels and the status of the immune cells, as well as the tumor infiltration of immune cells. In addition, RT-qPCR, Western blot, immunohistochemistry and several functional and animal experiments were performed for clinical, in vitro and in vivo validation. POLD1 was highly expressed in a variety of tumors including ccRCC, and further verified in a validation cohort of 60 ccRCC samples and in vitro cell line experiments. POLD1 expression levels in the ccRCC samples were associated with various clinical characteristics including pathologic tumor stage and histologic grade. ccRCC patients with high POLD1 expression have poor clinical outcomes and exhibit a higher rate of somatic mutations than those with low POLD1 expression. Cox regression analysis also showed that POLD1 could act as a potential independent prognostic biomarker. The DEGs associated with POLD1 were significantly enriched in the immunity-related pathways. Moreover, further immune infiltration analysis indicated that high POLD1 expression was associated with high NK CD56bright cells, Treg cells, and myeloid-derived suppressor cells’ (MDSCs) infiltration scores, as well as their marker gene sets of immune cell status. Meanwhile, POLD1 exhibited resistance to various drugs when highly expressed. Finally, the knockdown of POLD1 inhibited the proliferation and migration, and promoted the apoptosis of ccRCC cells in vitro and in vivo, as well as influenced the activation of oncogenic signaling. Our current study demonstrated that POLD1 is a potential prognostic biomarker for ccRCC patients. It might create a tumor immunosuppressive microenvironment and inhibit the susceptibility to ferroptosis leading to a poor prognosis.

## 1. Introduction

Renal cell carcinoma (RCC) is one of the common urological malignancies that accounts for approximately 3% of all adult malignancies. According to statistics, in 2020, there were more than 430,000 new cases of RCC diagnosed and approximately 180,000 cases of RCC-related deaths worldwide annually [1]. Since early renal cell carcinoma usually has no obvious clinical symptoms, with approximately 25–30% of patients having presented with metastases at the time of the initial diagnosis, it is associated with poor survival outcomes [2]. Despite considerable improvements in the diagnosis and treatment of RCC, the prognosis still remains challenging. The rapid development of high-throughput sequencing technology has enabled more driver genes to be successively discovered and validated [3,4,5]. Notably, the advent of immune checkpoint inhibitors (ICIs) has led to a paradigm change in the management of metastatic renal cell carcinoma (mRCC); nevertheless, the benefit of treatment is confined to a limited proportion of patients. Therefore, increasing numbers of researchers [6,7,8] are devoted to exploring the identification of predictive biomarkers of ICIs’ clinical benefit in order to increase the number of mRCC patients who may benefit from these treatments. Therefore, more effective key driver oncogenes are urgently needed to accurately predict the prognosis, especially those that affect the status of the immune microenvironment in RCC.

Located in 19q13.3-q13.4, POLD1 (DNA polymerase delta 1 catalytic subunit) is the encoding gene of the Polδ catalytic subunit, exhibiting both DNA polymerase activity and 3′ to 5′ exonuclease activity as well as playing a critical role in genomic copy with high fidelity, DNA damage repair processes, cell growth and differentiation [9]. Previous studies have shown that POLD1 is associated with the progression of various cancers. For example, the upregulation of POLD1 in breast and liver cancer is correlated with higher tumor mutation burden and poor prognosis [10,11,12]. In addition, the proofreading defect caused by inactivating mutations in the POLD1 proofreading (exonuclease) domain leads to a significant increase in somatic mutation load and cancer risk [13,14,15]. Meanwhile, recent studies have also indicated that the POLD1 proofreading domain mutation can potentially predict the clinical benefit in cancer patients that are treated with immune-checkpoint inhibitors (ICIs) [16,17]. All these findings have demonstrated the possibility of POLD1 as a biomarker for human cancers, and maybe a novel target in treating a variety of malignancies. However, the underlying mechanisms of POLD1 concerning tumor progression and immune engagement in RCC remain unknown.

In this present study, we evaluated the levels of POLD1 expression and its association with patients’ prognosis in a ccRCC cohort. We further explored the association of POLD1 expression with immune cell infiltration scores as well as marker gene sets of immune status via multiple databases. Meanwhile, we investigated the effect of POLD1 knockdown on the proliferation, apoptosis, migration and oncogenic signaling pathways of ccRCC cells. The final results pinpointed the vital role of the POLD1 gene in the prognosis of ccRCC and illustrated a relevance between POLD1 expression and the clinical outcome of the ccRCC cohort as well as the tumor-infiltrating immune cells.

## 2. Results

### 2.1. Upregulated POLD1 Expression in Tumor Samples Forebodes Poor Prognosis of ccRCC Patients

To compare the transcriptional expression of the POLD1 gene between tumor and non-tumor tissues, we first evaluated POLD1 expression in TCGA pan-cancer according to the TIMER online platform, and found higher POLD1 expression in various human tumors compared with their counterpart normal tissues (Figure 1A). Particularly, to explore the association between POLD1 expression and ccRCC patients, we used five GEO expression profiling by array RNA sequencing profiles from the TCGA and ICGC databases containing ccRCC tumor and non-tumor specimens. The results showed that the POLD1 gene expression level in the ccRCC samples were significantly increased in these four cohorts from the TCGA, ICGC, and GEO databases (Figure 1B–F). Meanwhile, a higher expression of POLD1 gene at the mRNA level was also investigated in our own validation cohort (60 pairs of ccRCC samples) (Figure 1G). Furthermore, high POLD1 protein expression was also observed in a validation cohort of 60 ccRCC samples (Figure 1H,I). Finally, the difference in POLD1 expression was also validated by the RT-qPCR and Western blot analysis of a normal human renal epithelial cell line and in RCC cell lines (Figure 1J). Together, these results highlighted a potential role for POLD1 as a regulator of ccRCC in tumorigenesis and progression.

To investigate the clinical prognosis of high-level POLD1 expression on the ccRCC cohort, we first divided the patients equally into two groups (POLD1^high^ and POLD1^low^) based on the median counts of POLD1 expression level. The results showed that significantly high POLD1 expression has a prominent clinical relevance with high-pathologic grade and clinical stage (T, N, and M stage) (Figure 2A–D). In addition, the association of POLD1 expression with clinicopathologic features was also assessed in the validation cohort (Appendix A). Furthermore, multiple survival analyses were also investigated for their association with POLD1 expression level in ccRCC patients. A survival analysis indicated that the OS, PFS, and DSS rates of ccRCC patients in the POLD1^high^ group were significantly lower than those with POLD1^low^ (Figure 2E–G). The same conclusion that POLD1 overexpression had poor OS was also evidenced in the validation cohort (Appendix A).

After univariate analysis, further multivariable Cox regression indicated that the POLD1 gene expression is an independent and significant risk prognostic factor for the OS in patients with ccRCC (Figure 2H,I). Although the univariate analysis also showed that POLD1 expression was associated with the OS of ccRCC patients in the validation cohort, the multivariate analysis demonstrated that POLD1 was not an independent indicator of poor OS (Appendix A). Based on the result of the multivariate Cox analyses with the TCGA cohort, the 3-, 5-, 8-year nomogram predicting the OS of ccRCC patients was established by age, grade, TNM stage and POLD1 mRNA expression (Figure 2J). As shown in the calibration curves (Figure 2K), the nomogram demonstrated satisfactory consistency. Notably, regional and distant metastases are a priori in this respect, unalterable but known “confounders” for the generation of prognostic models in early stage ccRCC. Thus, it would certainly be useful to focus primarily on localized RCC to limit cohort heterogeneity and prove POLD1 expression as an independent outcome variable with utility compared with traditional clinicopathologic variables. Therefore, we deleted the regional and distant metastases patients, leaving the localized ccRCC patients for further analysis of POLD1 expression as an additional independent outcome variable with utility compared with traditional clinicopathologic variables (Appendix A). In addition, a poorly justified dichotomization may lead to serious information losses. Thus, we supplemented extra data to explore the association between traditional clinicopathologic multiple categorical variables and POLD1 expression (Appendix A).

### 2.2. Genetic Alterations and Somatic Mutations Analysis with POLD1 Expression in ccRCC

We firstly investigated the mutation features of POLD1 in ccRCC from the TCGA cohort based on the cBioPortal tool. The results showed the genetic alteration frequency of POLD1 among TCGA pan-cancer including ccRCC (Appendix A); the mutation sites of POLD1 in ccRCC are displayed in Appendix A. As POLD1 plays a vital role in genomic copy with high fidelity and DNA damage repair processes, we further analyzed the association of the CNV and SNV of the POLD1 gene with the survival outcomes in ccRCC patients based on the GSCA. The finding showed that both the CNV and SNV of the POLD1 gene were statistically significant with the clinical outcomes of OS, PFS, and DSS in ccRCC patients (Figure 3A,B). Furthermore, we also obtained the mutation profiles from the TCGA-KIRC cohort to assess whether the distribution of mutations in the ccRCC cohort is influenced by the POLD1 gene level. We profiled the whole mutation atlas of the ccRCC patients in the POLD1^high^ and POLD1^low^ groups (Figure 3C), and found it different from the top five genes of mutations in the POLD1^low^ group (VHL (47%), PBRM1 (39%), TTN (16%), BAP1 (8%), MUC16 (7%)); the genes with a higher proportion of mutations in the POLD1^high^ group were VHL (43%), PBRM1 (40%), SETD2 (15%), TTN (15%), and BAP1 (11%) (Appendix A). In addition, we also noted that in the comparison of the mutations between the POLD1^high^ and POLD1^low^ groups, most genes had more mutations in the high POLD1 expression group (Figure 3D).

### 2.3. ccRCC Patients with POLD1 Differential Expression Showed Immunity-Related Characteristics

To further explore the underlying function of POLD1 in tumor tumorigenesis and progression, we assessed the DEGs in the TCGA-KIRC cohort between the POLD1^high^ and POLD1^low^ groups. The DEGs results include 2241 upregulated and 1202 downregulated genes (Figure 4A). Meanwhile, the top 20 significant upregulated and downregulated DEGs are shown in the single gene co-expression heat map (Figure 4B,C). KEGG enrichment and GO analyses were applied to detect the function of these DEGs. GO analyses in relation to these DEGs were primarily enriched in the regulation of leukocyte activation, the immune response-regulating signaling pathway, immunoglobulin complex, immunoglobulin receptor binding, and MHC protein complex binding (Figure 4D). These results demonstrated that most of the biological mechanisms were involved in the pathways or cellular biology of the immune response. KEGG enrichment analysis revealed several major pathways: pathogen infection-related pathways, Th17 cell differentiation, and antigen processing and presentation (Figure 4E). Notably, the Metascape database was also performed to pinpoint the underlying functional mechanisms markedly associated with the DEGs of POLD1 gene levels (POLD1^high^ vs. POLD1^low^). The results showed that the biological effects were also mainly enriched in the immune response (Figure 4F–H), which was consistent with the results of the KEGG and GO analyses.

### 2.4. The Association of POLD1 Expression with Marker Gene Sets of Immunoinfiltration and Immunoregulation

The above findings demonstrated that the high POLD1 expression level was related to rapid tumor progression. However, the underlying mechanisms of POLD1 regarding tumor progression and the immune microenvironment in RCC remained unknown. Therefore, we firstly tried to elucidate the relevance of POLD1 expression with the different biomarkers of TILs. After adjusting the association values for tumor purity, we found that POLD1 was strongly linked with the majority of TILs markers in ccRCC, including several functional T cells (Th1/Th2/Th17/Tfh cells, Tregs, and exhausted T cells) (Figure 5A). Particularly, there was a significant association between POLD1 gene and markers of Treg cells and T cell exhaustion, such as FOXP3, CCR8, STAT5B, PDCD1 (PD-1), CTLA4, and LAG3 (Figure 5A). This result intimated the close relationship between POLD1 levels and the T cells’ exhausted status, and that POLD1 might play a vital role in immune escape in the microenvironment of RCC. In addition, the markers of TAMs, M1 macrophages, M2 macrophages, monocytes, and neutrophils also had a significant association with the POLD1 level (Figure 5A). We also validated the association between POLD1 levels and the aforementioned markers of Treg cells and T cell exhaustion via the GEPIA database, and the associations were consistent with those conducted in TIMER (Appendix A).

Immunomodulators are one of the crucial factors that regulate the underlying function of the immune system. The findings suggested that POLD1 was significantly correlated with immunoinhibitors, including CTLA4, LAG3, LGALS9, TGFB1, and PDCD1 (PD-1) (Figure 5B). The expression of POLD1 was also strongly linked with immunostimulators, such as TNFRSF18, TNFRSF25, TNFRSF8, TNFRSF14, and LTA (Figure 5B). In addition, chemokines (or receptors) play an important function in regulating the infiltration degree of immune cells. This research implicated the significant association between POLD1 expression and chemokines, including CCL5, CXCL13, XCL1, and XCL2 (Figure 5C), as well as the chemokine receptors containing CXCR3, CXCR5, and CXCR6 (Figure 5C). Therefore, we suspected that POLD1 may function as an immunoregulatory factor in ccRCC.

In summary, these results demonstrated that POLD1 is potentially engaged in the regulation of immune interactions and may assist tumor immune escape.

### 2.5. Characteristics of Immune Cell Infiltration in ccRCC Patients with POLD1 Differential Expression

Immune infiltration is the vital factor associated with tumor progression. Although we had investigated the association of POLD1 expression with markers of immunoinfiltration and immunoregulation, the characteristics of immune cell infiltration in ccRCC patients with POLD1 differential expression level remained unknown. Therefore, we next assessed the relationship of POLD1 levels with the infiltrating abundance of various immune cells based on different databases (TIMER and TISIDB). The results of both algorithms showed that POLD1 levels were positively related with the infiltration of CD56dim natural killer cell, regulatory T cell (Treg), myeloid-derived suppressor cells (MDSC), and activated CD8 T cells in ccRCC (Figure 6A,B). However, POLD1 levels showed a negative association with the infiltration of immature dendritic cells, natural killer cells, and neutrophils (Figure 6A,B). Next, to further explore the composition of 22 immune cells in the ccRCC tumor immune microenvironment (TME) in different POLD1 gene levels, the CIBERSORT algorithm was applied to dissect the TME of ccRCC patients (POLD1^high^ vs. POLD1^low^ groups). The final results showed that CD4+ T cells (memory resting), macrophages M2, CD8+ T cells, and macrophages M1 were the top four proportions of immune cells (Figure 6C). The TME of tumors with high POLD1 expression was indicated to be suppressive, as shown in the higher infiltration of Treg cells and the relatively low infiltration of CD8+ T cells (Figure 6C). Meanwhile, these 22 immune cells are divided into four categories, namely dendritic cells, lymphocytes, macrophages, and mast cells. We also analyzed the differences in infiltration of four categories between the POLD1^high^ and POLD1^low^ groups. Regrettably, the proportions of the four categories did not show statistically significant differences between the two groups (Appendix A). Notably, as the TME scores could reflect the abundance of immune and stromal elements in TME, we further used the ESTIMATE algorithm to calculate the TME scores between the POLD1^high^ and POLD1^low^ groups, including the stromal score, immune score, and estimate score. The findings showed that patients in the POLD1^high^ group had higher TME scores (Figure 6D). Furthermore, the ssGSEA algorithm was used to predict the abundant difference of 28 TIICs between the POLD1^high^ and POLD1^low^ groups (Appendix A). Interestingly, the high-POLD1 expression group showed higher immune enrichment scores (Figure 6E). Among them, the immune-suppressive activity cells, such as MDSC and Treg cells, were higher than that in tumors with low POLD1 expression (Figure 6E). Notably, the activities of the 28 TIICs were divided into two categories (anti-tumor immunity and pro-tumor, immune suppressive functions) according to the function of the immune cells in the TME [18]. Spearman’s correlation analysis indicated the heterogeneity among the POLD1^high^ and POLD1^low^ groups in promoting tumor activity and anti-tumor activity (Figure 6F). Moreover, the three critical immune checkpoints (PD-1, PD-L1, and CTLA-4), especially the expression of PD-1 and CTLA-4, were notably higher in the POLD1^high^ than the POLD1^low^ group (Appendix A).

### 2.6. Association between POLD1 and Anticancer Drug Sensitivity

In order to assess the probable benefit to patients for ICI therapy in different POLD1 expression groups, the TIDE-score was used to assess the potential clinical efficacy of immunotherapy in different POLD1 expression groups. In our results, the POLD1^high^ group had a higher TIDE score than the POLD1^low^ group (Figure 7A). In addition, we found that the POLD1^high^ group had a lower microsatellite instability (MSI) score, while the POLD1^high^ group had a higher T cell dysfunction score; however, there was no difference in T cell exclusion between the two groups. In addition, we also assessed the association between POLD1 gene expression and GDSC/CTRP anticancer drug sensitivity via the Spearman’s correlation. The findings demonstrated that a positive correlation implicated that high POLD1 expression confers resistance to the drug and a negative one does not. As the results show in Figure 7, POLD1 exhibited the potential resistance to multiple drugs when highly expressed. Taken together, the expression of POLD1 may have a relationship with the efficacy of various anticancer drugs, which might be another potential prognostic factor for ccRCC patients.

### 2.7. POLD1 Knockdown Inhibited the Malignant Biological Behaviors and Signaling of ccRCC Cells In Vitro and In Vivo

We further explored the role of POLD1 in ccRCC in vitro. The interference efficiency of POLD1 knockdown by siRNA was validated using RT-qPCR and Western blotting in 786-O and A498 cells (Figure 8A,B). The results of the CCK-8, EdU, and colony formation assays all showed that the knockdown of POLD1 significantly inhibited the proliferation ability of the 786-O and A498 cells (Figure 8C–E). Further, the flow cytometry results showed that a significantly higher proportion of apoptotic cells in POLD1 knockdown ccRCC cells, and the Western blot also detected a marked increase in the apoptosis-related protein PARP and Caspase-3 spliceosomes (Figure 8F,G). The above results verified that the knockdown of POLD1 promoted the ccRCC cells’ apoptosis. In addition, POLD1 knockdown also significantly inhibited the migratory ability of the ccRCC cells, as evidenced via the Transwell and wound healing assays (Figure 8H,I). To further explore the mechanisms through which POLD1 regulates the ccRCC anti-tumor immune response, we detected the impact of POLD1 on several oncogenic signaling pathways using a Western blot. The results demonstrated that POLD1 knockdown in 786-O and A498 cells significantly decreased the levels of autophagy and ferroptosis signaling pathway-related genes, including p62, NRF2, Keap1 and GPX4 (Figure 9A). Finally, depletion of POLD1 also inhibited tumor growth in the xenograft mouse model (Figure 9B–D). Taken together, POLD1 facilitated the proliferation and migration, and inhibited the apoptosis in ccRCC cells in vitro and in vivo, as well as influenced the activation of oncogenic signaling.

## 3. Discussion

Accurate and efficient DNA replication is a critical process occurring in cell proliferation and ensures the faithful and timely transmission of genetic information. The DNA damage response pathway is coordinated with DNA repair, cell cycle progression and proliferation. Notably, DNA polymerase δ works in both DNA replication and repair processes [19]. POLD1 (DNA polymerase delta 1 catalytic subunit) is the encoding gene of the Polδ catalytic subunit that exhibits both in DNA polymerase activity and 3′ to 5′ exonuclease activity; it plays a critical role in genomic copy with high fidelity, DNA damage repair processes, cell growth and differentiation [9,20]. The proofreading defect caused by inactivating mutations in the POLD1 proofreading (exonuclease) domain leads to a significant increase in somatic mutation load and cancer risk [13,14]. Previous studies have suggested that the alterations in POLD1 in humans are associated with multiple cancers, including hepatocellular carcinoma, colorectal cancer, endometrial cancer, and breast cancer [11,14,15]. Meanwhile, POLD1 upregulation contributes to cancer cell proliferation, migration, and invasion, which may be attributed to surviving replication stress via improving their tolerance to DNA damage [21,22]. Furthermore, recent studies have also indicated that POLD1 proofreading domain mutation can potentially predict the clinical benefit in cancer patients who are treated with immune-checkpoint inhibitors (ICIs) [16,17]. Therefore, POLD1 may play an essential role in cell progression and tumorigenesis. However, the relationship between POLD1 and ccRCC remains unknown. In this study, we systematically characterized POLD1 in ccRCC, revealing its expression profile, diagnostic and prognostic significance, somatic mutation difference, potential functions, immune infiltration, anticancer drug sensitivity prediction, and its association with the infiltration levels of immune cells. Together, these observations strongly revealed new insights in understanding the potential function of POLD1, and it may serve as a prognostic biomarker correlated with immune infiltration in ccRCC.

We first evaluated POLD1 expression in TCGA pan-cancer according to the TIMER online platform, and found higher POLD1 expression in various human tumors compared with their counterpart normal tissues. Similarly, elevated POLD1 expression has been verified via independent researches in several cancers, including hepatocellular carcinoma, breast cancer, and colorectal cancer [10,11]. The higher transcriptional levels of POLD1 were also observed in ccRCC tissues from clinical samples as well as the TCGA, ICGC and GEO databases. Meanwhile, POLD1 mRNA and protein expression levels in ccRCC tissues were significantly higher than that in normal renal tissues in our own validation cohort, which was also verified in other independent research [23]. To further verify our conclusion, we detected the relative expression levels of POLD1 in various RCC cell lines and a normal renal cell line using RT-qPCR and Western blotting. The results turned out to be consistent with our bioinformatic analyses.

We further analyzed the relationship between POLD1 expression and the clinical characteristics of ccRCC patients, revealing that POLD1 expression was significantly associated with pathologic tumor stage and histologic grade. The KM analysis demonstrated that overexpression of POLD1 was suggestive of the poor OS, PFS, and DSS prognoses of ccRCC patients. Univariate and multivariate regression analyses confirmed that POLD1 was an independent undesirable prognostic factor for OS in ccRCC. Based on the above results, we further constructed a prognostic nomogram by integrating several clinical parameters and POLD1 expression from the TCGA-KIRC cohort to predict individual patient mortality risk (3-, 5-, and 8-year) and assist in optimizing treatment decisions.

The impacts of POLD1 mutations have been extensively researched in multiple cancers [13]. We also explored the genetic alterations and somatic mutations association with POLD1 expression in ccRCC. Although the mutation frequencies of POLD1 in ccRCC was not high compared with other cancers, and high tumor mutational burdens were not observed in ccRCC, the findings showed that both the CNV and SNV of the POLD1 gene were statistically significant with the clinical outcomes of the OS, PFS, and DSS in ccRCC patients. Furthermore, a difference in the mutational spectrum was observed between the POLD1^high^ and POLD1^low^ groups. Several genes associated with growth and development (LRP2, LAMC2, PCDH15) and oncogenes (CASP8AP2, POLR2A, SETD2) had higher mutation rates in the POLD1^high^ group.

To investigate the abnormal changes in the downstream pathways caused by the differential POLD1 expression in ccRCC, we identified the DEGs between the ccRCC patients in the POLD1^high^ and POLD1^low^ groups. KEGG enrichment and GO analyses results showed that the above DEGs mainly participated in the regulation of leukocyte activation, the immune response-regulating signaling pathway, immunoglobulin complex, immunoglobulin receptor binding and MHC protein complex binding. Furthermore, the Metascape database also suggested that the biological effects were mainly enriched in the immune response. All the above results indicated that altered POLD1 expression may play an unexpected role in anti-tumor immune response regulation.

The immunogenicity of a tumor depends on the tumor’s antigens and the abundance of immune cell infiltration, as well as the type and quantity of the immunomodulatory molecules presenting within the tumor’s microenvironment [24]. As one of the major components of a tumor’s microenvironment, immune cell infiltration has been shown to play a critical role in the development and progression of cancers [25,26,27]. Renal cell carcinomas are characterized by abundant leukocyte infiltration, including CD8^+^ T cells, CD4^+^ T cells and natural killer (NK) cells, as well as myeloid cells with characteristics of macrophages and neutrophils [28,29]. Notably, this study discovered that POLD1 is strongly related to the abundance of immune infiltration in ccRCC. We speculated that the high expression of POLD1 could perturbate the tumor microenvironment of RCC. The findings showed that POLD1 expression had a strong association with TILs including NK CD56bright cells, Treg cells, CD8+ T cells, and MDSCs. Among them, CD56bright NK cells are probably believed to be the immediate precursors of CD56dim NK cells. Unlike CD56dim NK cells with strong cytotoxicity, CD56bright NK cells have stronger cytokine secretion functions, despite having only weak antibody-dependent cellular cytotoxicity [30]. It has been reported that the expression pattern of the surface-activated receptors and inhibitory receptors in NK cells could alter with the changing tumor microenvironment. For example, directing the generation of CD56bright NK cells with an immature phenotype expressing more inhibitory receptors would dampen the cytotoxicity of NK cells and further lead to the reduction in NK cell-mediated tumor killing [31,32]. Based on our study showing the positive association of POLD1 expression and CD56bright NK cell infiltration and the negative association with CD56dim NK cells, we hypothesized that the relatively higher POLD1 expression in ccRCC may imply a higher proportion of CD56bright NK cells and a stronger immunosuppressive environment. Meanwhile, a higher proportion of Treg cell infiltration in tumors with high POLD1 expression may also support this hypothesis. Treg cells are vital for maintaining T-cell tolerance to self-antigens. Recent evidence has demonstrated that Treg cells are thought to dampen T-cell immunity to tumour-associated antigens and to be the main obstacle tempering successful tumor immunotherapy [33,34,35]. Paradoxically, an abundance of intratumoral CD4+ and CD8+ T cells were related to a high tumor grade and shorter survival of the patients. However, the high proliferation rate of tumor-infiltrating CD8+ T cells was associated with prolonged patient survival in ccRCC. This may be due to the massive depletion of tumor-infiltrating T cells, leaving behind the infiltration of tumor tissue by CD8(+) T cells bearing proliferative activity could eventually work as anti-tumor effector cells [36]. In addition, elevated levels of MDSCs are also found to be related to tumor progression in patients with various cancers including ccRCC [37,38,39]. Based on the analysis of circulating MDSCs, monocytic mononuclear MDSCs (M-MDSCs) were initially reported to be primarily immunosuppressive, whereas granulocytic polymorphonuclear MDSCs (PMN-MDSCs) were primarily pro-angiogenic [40]. Nevertheless, subsequent studies demonstrated that immunosuppressive and angiogenic molecules were generated by the both subcategories and especially by those MDSCs that infiltrate within the TME [41,42]. Among them, MDSCs are known to exert their immunosuppressive effects through multiple mechanisms, including the generation of arginase 1, TGFβ and IL-10, nitrosylation of the TCR, downregulation of CD62L, and cysteine sequestration [43,44,45,46,47,48,49]. Similarly, overexpression of POLD1 has a strong association with MDSCs. Therefore, as the mobilization and recruitment of MDSCs are dependent on various tumor-derived factors, therapeutic interventions that interrupt either these molecules or the MDSC effectors themselves could substantially change the response to immunotherapy [49].

Furthermore, upregulated POLD1 expression was not only strongly correlated with Treg cell markers (FOXP3, CCR8) and T-cell exhaustion markers (PD1, CTLA4, LAG3) but also associated with the cell’s response to chemokines. Treg cells mediate immunosuppression through multiple pathways to facilitate tumor immune-evasion. FOXP3 is one of the crucial markers of Treg cells and serves as a necessary transcription factor for its immunosuppressive function. FOXP3+ Treg cells have been found to be significantly elevated in peripheral blood as well as tumor tissues, and upregulated infiltration predicted poor prognosis in a variety of cancers, including breast cancer, melanoma, non-small cell lung cancer, and gastric cancer [50,51]. CCR8 is one of the members of the chemokine receptor subfamily that marks highly suppressive Treg cells within tumors and actively participates in the recruitment of Treg cells and Th2 cells to tumors [52]. Immune checkpoint inhibitor (ICI) therapy is the primary immunotherapeutic strategy. As the crucial immune checkpoint component, PD1/PDL1 have been verified to modulate the function of TILs. To date, ICI therapy targeting the PD1/PDL1 checkpoints has been widely applied to a variety of malignancies including ccRCC [53,54,55]. Several studies have found that ICIs (including PD-1/PD-L1/CTLA4/LAG3) have an important function in maintaining tumor antigen tolerance but are usually hijacked by tumors to mediate their immune escape, leading to poor therapeutic outcomes in some patients with ICI therapy [56,57]. Therefore, a better understanding of the mechanisms underlying the responses or resistance to immune checkpoint inhibitors and cytokines is vital for improving the clinical efficacy of this therapeutic modality. In conjunction with these results, POLD1 played a crucial function in recruiting and modulating TILs in ccRCC; and the molecular mechanism and function of POLD1 in regulating the tumor microenvironment also pinpoints the direction for our future research.

It was notable that Godlewski et al. [23] also examined POLD1 expression in ccRCC but found increased POLD1 expression to be a favorable marker for overall survival, which was contrary to our current results. In our results, POLD1 expression levels were significantly higher in the ccRCC tissues compared to the counterpart/paired peritumoral normal renal tissues in various ccRCC cohorts, including the TCGA-KIRC cohort, GEO cohort (GSE46699 and GSE53757), E-MTAB-1980 cohort, and the 60 ccRCC validation cohort. Among them, POLD1 may be considered as an unfavorable prognostic biomarker for overall survival in the TCGA-KIRC cohort and E-MTAB-1980 cohort. In addition, several in vitro functional and in vivo animal experiments also demonstrated that POLD1 may serve as an oncogene in ccRCC. Furthermore, a recent study on ccRCC also disclosed that POLD1 is the hub protein in the protein–protein interaction networks related to ccRCC progression [58]. Our current preliminary mechanism exploration indicated that POLD1 may create a tumor immunosuppressive microenvironment and inhibit the susceptibility to ferroptosis leading to a poor prognosis. Taken together, the prognostic and clinicopathological significance of POLD1 among the ccRCC patients remains apparently controversial. It is necessary to further explore the exact molecular mechanism on the role of POLD1 in ccRCC. We will also include more ccRCC validation cohorts and assess the postoperative follow-up data for analyzing and validating the significance of the POLD1 protein as a possible prognostic marker of patient survival in the future.

Our enrichment analysis results showed that POLD1 is closely related to immunity-related characteristics, especially anti-tumor immune response regulation. Further analyses indicated that POLD1 knockdown inhibited proliferation and migration, and facilitated apoptosis in ccRCC cells in vitro and in vivo. In addition, POLD1 knockdown in ccRCC cells significantly decreased the levels of autophagy and ferroptosis signaling pathway-related genes, including p62, NRF2, Keap1 and GPX4, which were reported to serve as the vital regulatory factors in many types of cancers [59,60,61]. These results suggested that POLD1 knockdown may enhance the susceptibility of tumor cells to autophagy/ferroptosis through certain pathways. It is noteworthy that increasing studies have systematically indicated that ferroptosis could generate multiple effects on regulating tumor immune tolerance. On one hand, ferroptosis could mediate tumor peripheral immune tolerance via the co-stimulatory, co-inhibitory and checkpoint pathways [62,63,64,65]. On the other hand, ferroptosis of tumor cells could promote tumor immune escape by activating the functions of different immune cells including CD8+ T [66], Tregs [67], MDSCs [68] and NK cells [69] in the immune microenvironment. Hence, we speculate that the ferroptosis-related pathways resistance of tumor cells was downregulated, and the susceptibility to ferroptosis was increased after POLD1 knockdown, thus, promoting the immune tolerance of tumor cells, which indicates that ferroptosis might represent a new curative option for treating immunotolerant cancers. Nevertheless, a comprehensive exploration of the mechanisms behind how to modulate tumor immune tolerance by inducing or suppressing cellular ferroptosis is essential for future investigations.

There exist several limitations in our study, although we explored the prognostic value and underlying mechanisms of POLD1 in the development and progression of ccRCC from different perspectives. A limitation of this study is that the expression of POLD1 and its prognostic significance was based on preexisting data from the various cohorts of databases; although some of the findings have been verified by our limited validation cohort, more clinical samples will need to be validated to be conclusive. Secondly, most data in online databases are updated and expanded continuously, which may affect the final results of research. Thirdly, the biological functional assessment of POLD1 was based on in vitro experiments and only one cell line was used for the in vivo experiments; furthermore, it is necessary to explore in vivo tumor model experiments to verify the above findings. Finally, although this study has demonstrated the potential role of POLD1 in tumor immune regulation and several oncogenic signaling pathways, the autophagy and ferroptosis signaling pathways are both complex processes. Further in-depth research on the specific molecular mechanisms of how POLD1 is involved in the autophagy and ferroptosis signaling pathways and the relationships with immunoregulation pinpoint the direction for our future work.

## 4. Materials and Methods

### 4.1. Datasets Sources and Ethics Statement

We downloaded the data (including the mRNA expression profiles and clinicopathological characteristics) from the kidney renal clear cell carcinoma (KIRC) dataset in The Cancer Genome Atlas (TCGA) database https://portal.gdc.cancer.gov/, accessed on 1 June 2022, which was comprised of 539 KIRC samples and 72 paired paracancerous control tissues (Workflow Type: STAR-Counts). A summary of the clinical data is shown in Appendix A. The TPM values (transcript per million) were selected to compare differential expression among the KIRC samples. The RNA-seq data of the KIRC-CN cohort contains 90 primary tumor tissues and 45 non-tumor specimens, which were downloaded from the International Cancer Genome Consortium (ICGC) Data Portal https://dcc.icgc.org/, accessed on 1 June 2022. Transcriptional profiles of two independent cohorts (GSE46699 and GSE53757), containing tumor and non-tumor tissues, were also extracted from the Gene Expression Omnibus (GEO) database https://www.ncbi.nlm.nih.gov/geo/, accessed on 2 June 2022. In addition, matrix files of gene expression profiles and the clinical information of the E-MTAB-1980 cohort were downloaded from the ArrayExpress website https://www.ebi.ac.uk/arrayexpress/experiments/E-MTAB-1980/, accessed on 2 June 2022, and the study of Sato et al. [70]. The following clinical exclusion criteria were identified: (i) missing or incomplete cause of death; (ii) duplicate patients; (iii) survival times < 30 days; and (iv) age < 18 years. All analyses were consistent with the Helsinki Declaration (as revised in 2013).

We randomly selected tumor and paired paracancerous kidney tissues from 60 ccRCC patients undergoing radical nephrectomy. The samples were acquired from the First Affiliated Hospital, School of Medicine, Zhejiang University, between January 2020 and December 2022 (Appendix A). This study strictly followed the principles of medical ethics and was authorized and supervised under the Institutional Ethics Committee of the First Affiliated Hospital of Zhejiang University (ID: IIT20200733A, approval date 24 September 2020).

### 4.2. Differential Expression Analysis

POLD1 mRNA expression data in pan-cancer analysis were retrieved from TIMER http://timer.cistrome.org/, accessed on 10 June 2022, a handy tool used to explore TCGA datasets. The “limma” package [71] in R software (version 4.1.3) was applied to evaluate POLD1 differentially expressed between tumor and non-tumor tissues in ccRCC. The threshold parameters were set as |log2 Fold Change| > 1.5 and *p* value < 0.05. To verify the protein expression level of POLD1, immunohistochemical results of POLD1 in ccRCC tissues and paracancerous normal kidney tissues were performed from the validation cohort of 60 ccRCC samples.

### 4.3. Prognostic Significance Analysis

The association between POLD1 expression and clinicopathological features, including the ccRCC patient’s age, gender, individual cancer stage, metastasis status (regional lymph nodes and distant) and tumor histology grade, was conducted by the TCGA-KIRC and ArrayExpress (E-MTAB-1980) cohorts. Meanwhile, the survival outcomes of POLD1 in ccRCC, including the overall survival (OS), progression-free survival (PFS) and disease-specific survival (DSS), were investigated by Kaplan–Meier (KM) plotter algorithm. In addition, univariate and multivariate Cox regression analyses were performed to determine the prognostic factors of the ccRCC patients, which were further applied to the nomogram model analysis for prognostic evaluation.

### 4.4. Functional Enrichment Analysis

We obtained the differentially expressed genes (DEGs) between the POLD1^high^ and POLD1^low^ expression groups identified based on the median counts of POLD1 expression by using the “limma” package. The “ggplot2” package in R was employed to visualize volcano plots and heat maps of the DEGs. To functionally annotate those DEGs, gene ontology (GO) annotations and Kyoto Encyclopedia of Genes and Genomes (KEGG) pathways functional analyses were further explored and visualized via DAVID https://david.ncifcrf.gov/, accessed on 12 June 2022 (v2022q3) and Metascape https://metascape.org/, accessed on 12 June 2022. Meanwhile, the enriched pathways were also visualized using the “tidyr” and “ggplot2” packages of R software (version 4.1.3).

### 4.5. Somatic Mutation Analysis

Firstly, we explored the POLD1 alteration frequency, copy number alteration (CNA) and mutation type in TCGA PanCancer Altas via the cBioPortal http://cbioportal.org/, accessed on 20 June 2022 for the Cancer Genomics database. Secondly, we used the Gene Set Cancer Analysis (GSCA) http://bioinfo.life.hust.edu.cn/GSCA/, accessed on 20 June 2022 [72] integrated platform to investigate the association of the Copy number variation (CNV) and Single Nucleotide Variation (SNV) of POLD1 with the survival outcomes (OS, PFS, and DSS) in ccRCC patients. Finally, there were 357 ccRCC cases with mutation profiles stored in the TCGA-GDC database. We divided the mutation data into two groups (POLD1^high^ and POLD1^low^ group) according to the original POLD1 transcriptomic expression levels. The whole and respective mutation maps of the POLD1^high^ and POLD1^low^ groups were visualized by using the “maftools” package (“oncoplot” function), and the distribution of significantly different mutant genes were evaluated via the “mafCompare” function in the “maftools” package.

### 4.6. Tumor Immunoinfiltration and Immunoregulation Analysis

TISIDB http://cis.hku.hk/TISIDB/, accessed on 20 June 2022 and TIMER http://timer.cistrome.org/, accessed on 20 June 2022 are the online platforms that enable analyses of the immune cell-tumor interactions for facilitating comprehensive study. We use these two different online platforms to calculate the relationships between POLD1 expression and the tumor immune cell. Meanwhile, TISIDB was utilized to analyze the association of POLD1 with immunostimulators, immunoinhibitors, and chemokines (or receptors) in ccRCC. In addition, the association of gene markers of tumor-infiltrating lymphocytes (TILs) with mRNA expression of POLD1 was also investigated on the TIMER website. To further describe the characteristics of the immune cells, the TCGA-KIRC gene expression profiles were analyzed using CIBERSORT [73] and single-sample gene set enrichment analysis (ssGSEA) [74]. The proportions of 22 tumor-infiltrating immune cells (TIICs) from two defined groups (POLD1^high^ and POLD1^low^ groups) were determined by using the “CIBERSORT” package. The ssGSEA algorithm in the “GSVA” package was used to predict the abundant difference of 28 TIICs between the POLD1^high^ and POLD1^low^ groups.

### 4.7. Drug-Sensitivity Analysis

Jiang et al. [75] indicated that the Tumor Immune Dysfunction and Exclusion (TIDE) score could predict the patient’s response to immunotherapy as it can reflect the potential capacity for the tumor’s immune evasion; a higher TIDE score was associated with poorer ICI efficacy. Therefore, the TIDE score of ccRCC patients among the POLD1^high^ and POLD1^low^ groups was calculated online http://tide.dfci.harvard.edu/, accessed on 25 June 2022. In addition, in order to investigate the association between POLD1 expression and drug sensitivity, we acquired the area under the dose-response curve values of various drugs and the corresponding gene-expression profiles of all cancer cell lines from the Cancer Therapeutics Response Portal (CTRP) [76] and the Genomics of Drug Sensitivity in Cancer (GDSC) [77] database. Spearman’s correlation analysis was performed to assess the association between POLD1 gene expression and GDSC/CTRP drug sensitivity. The findings suggested that negative gene expression does not confer resistance to the drug while a high gene expression does.

### 4.8. Cell Culture, Transfection, and Infection

A normal human renal epithelial cell line (HK2) and RCC cell lines (786-O, 769-P, ACHN, Caki-1, and A498) were purchased from China Cell Bank (Shanghai, China), with an STR identification certificate. All cell lines were cultured in Dulbecco’s modified Eagle’s medium (DMEM) or RPMI Medium 1640 basic (Gibco, New York, NY, USA) containing 10% fetal bovine serum (FBS) (ExCell Bio, Shanghai, China) and 1% penicillin/streptomycin (Fdbio science, cat#FD7016) at 37 °C with 5% CO_2_.

Cell transfection was performed using the jetPRIME^®^ Transfection Reagent (Polyplus) according to the manufacturer’s protocol. Briefly, cells were plated in six-well plates (NEST) and grown to 30–50% confluency (adherent cells) and then transfected and cultured at 37 °C for a further 48 h, followed by testing for reverse-transcription quantitative polymerase chain reaction (RT-qPCR), Western blotting (WB), and other experiments. POLD1 siRNA oligonucleotides (50 uM) siPOLD1#1 (forward 5′-GCUUCGCUCCCUACUUCUACATT-3′ and reverse 5′-UGUAGAAGUAGGGAGCGAAGCTT-3′), siPOLD1#2 (forward 5′-CGGGACCAGGGAGAAUUAAUATT-3′ and reverse 5′-UAUUAAUUCUCCCUGGUCCCGTT-3′), siNC oligonucleotides (forward 5′-UUCUCCGAACGUGUCACGUdTdT-3′ and reverse 5′-ACGUGACACGUUCGGAGAAdTdT-3′) were obtained from SUNYA (Hangzhou, China).

Lentiviruses expressing scrambled shRNA and shPOLD1 (forward 5′-CCGGCCTGGCACTGATGGAGGAGATCTCGAGATCTCCTCCATCAGTGCCAGGTTTTTG-3′ and reverse 5′-AATTCAAAAACCTGGCACTGATGGAGGAGATCTCGAGATCTCCTCCATCAGTGCCAGG-3′) were purchased from Genechem (Shanghai, China). The 786-O cells were used to establish stable POLD1 knockdown models. In total, 10^5^ cells were plated into a 6-well plate and transfected with indicated lentivirus according to the instructions. Then, 2 ug/mL (MCE, Austin, TX, USA) puromycin was used for screening infected cells for 1 week or more, with the transfection efficiency determined by RT-qPCR and Western blotting analysis.

### 4.9. RNA Extraction and Quantitative Real-Time PCR

Total RNA was extracted from cultured cells and tissue samples using Trizol reagent (life technologies, ambion^®^, Austin, TX, USA). ChamQ Universal SYBR qPCR Master Mix (Vazyme, Nanjing, China) with a PCR detection system (Bio-Rad, CFX96^TM^, Hercules, CA, USA) was applied to investigate the target genes’ expression. cDNA was synthesized using the HiScript^®^ II Q RT SuperMix for qPCR (Vazyme, Nanjing, China). Transcriptional levels were normalized against those of the internal control gene GAPDH. The following primers were listed: GAPDH forward 5′-GGAGCGAGATCCCTCCAAAAT-3′ and reverse 5′-GGCTGTTGTCATACTTCTCATGG-3′; POLD1 forward 5′-AGCAGGTCAAGGTCGTATCC-3′ and reverse 5′-AGCGTGGTGTAACACAGGTTG-3′. The 2^-ΔΔCT^ calculation method was applied to determine the relative target gene level.

### 4.10. Western Blotting

Protease and phosphatase inhibitors were added to lyse cultured cells in a radioimmunoprecipitation analysis (RIPA) solution. The BCA protein assay kit (Fdbio science, cat#FD2001) was used to determine the protein concentration. Western blot analysis was performed as previously described [78]. The Enhanced Chemiluminescent detection kit (Fdbio science, Hangzhou, China) was used to visualize the protein bands. The following antibodies were applied: anti-POLD1 (Abclonal, cat#A5323), anti-GAPDH (Fdbio science, cat#FD0063), anti-Parp (CST, cat#9542), anti-Caspase3 (CST, cat#14220), anti-cleave-Caspase3 (CST, cat#9664), anti-p62 (CST, cat#39749), anti-NRF2 (CST, cat#12721), anti-Keap1 (CST, cat#8047), anti-GPX4 (CST, cat#59735).

### 4.11. Immunohistochemistry

The paraffin-embedded ccRCC samples after surgery were subjected to immunohistochemistry (IHC) staining using the two-step method of the Dako Envision™ Detection System (DakoCytomation, Glostrup, Denmark). Primary antibodies anti-POLD1 (Proteintech, cat#15646-1) were applied. Images were captured using a fluorescence microscope (Olympus, Tokyo, Japan). IHC score was calculated by multiplying the staining intensity score by the positive rate score. A staining intensity score of 0, 1, 2 or 3 represented no-staining, weak-staining, moderate-staining or strong-staining, respectively. As well as the positive rate score, 0, 1, 2, 3 or 4 implied positive areas of 0–5%, 6–25%, 26–50%, 51–75% or 76–100%, respectively.

### 4.12. Cell Proliferation, Apoptosis, and Migration Assays

Cell proliferation was investigated using Cell Counting Kit-8 (CCK-8), 5-ethynyl-2′-deoxyuridine (EDU) and colony formation experiments. For the CCK-8 assay, 2500 cancer cells were seeded into each well of a 96-well plate and cultured at 37 °C in a cell incubator. CCK-8 reagent (MCE^®^ MedChem Express, Monmouth Junction, NJ, USA) diluted in the culture medium was added to the wells according to the manufacturer’s protocol at the same time point of each day. Absorbance values at 450nm were detected using a microplate reader (BioTek, Synergy Neo2, Winooski, VT, USA). For the EdU assay, 100 uL medium containing 50 µM EdU (UElandy, Suzhou, China) was add into each well of a 96-well plate for 2 h. Then, the cells were fixed with 4% polyformaldehyde for 20 min and subsequently incubated with a YF^®^ 488 Azide staining solution and Hoechst 33,342 for 30 min. Nuclei of all cells were stained with Hoechst 33,342 (blue) and proliferating cells were stained with YF^®^ 488 Azide (green). The stained cells were observed with a fluorescence microscope (Olympus, Tokyo, Japan) and photographed. Image J was applied to count the number of all cells and proliferating cells for calculating the proliferation rates. For the colony formation assay, 3000 cancer cells were cultured to each well of six-well plates and, changing the culture medium every other day for 1 week, followed by the fixation with 4% paraformaldehyde and afterwards stained with crystal violet.

Cell apoptosis was detected using ta flow cytometer. The transfected ccRCC cells were transferred to polystyrene FACS tubes for double staining with Annexin V-FITC/propidium iodide (PI) according to the manufacturer’s protocol (Annexin V-FITC/PI Apoptosis Kit; MultiSciences, Hangzhou, China). Flow cytometry (Beckman, Brea, CA, USA) was applied to detect the differences in the percentages of stained cells.

Cell migration ability was explored using wound healing and Transwell experiments. For the wound healing assay, a 200 mL pipette tip was applied to artificially make a single wound in each well when the confluence of transfected cells reached 90% in the six-well plate. The cell wound healing distance was assessed every 12 h after being changed to serum-free medium. The migration assay was completed in an 8 mm Transwell chamber (Corning, Costar^®^, Washington, DC, USA). Transfected ccRCC cells were inoculated on the upper chamber containing serum-free medium at an approximate density of 30,000 cells. To the lower chamber, 600 mL medium containing 30% FBS was added. After incubation at 37 °C for 48 h, invasive cells were immobilized with 4% paraformaldehyde and stained with crystal violet.

### 4.13. Animal Experiments

BALB/c nude mice (4–5 weeks, female) were used for the in vivo tumor model. All animal experiments were approved by the Institutional Animal Care and Use Committee of The First Affiliated Hospital, School of Medicine, Zhejiang University (ST2023006, approval date 20 January 2023). For the subcutaneous tumor model, 786-O cells (1 × 10^6^) transfected with scrambled shRNA and POLD1 shRNA and resuspended in 200 uL PBS were inoculated subcutaneously into the lower right flank regions of the nude mice. The width (W) and length (L) of the subcutaneous tumors were measured every week, and the volume (V) of each tumor was estimated as follows: V = (W^2^ × L/2). At the end of feeding (4 weeks or more), the mice were sacrificed with the tumors removed to record the weight and to take pictures.

### 4.14. Statistical Analysis

Student’s *t*-test or Mann–Whitney U-test were applied to calculate the differences between the transcriptional levels of the two defined groups. The association of POLD1 levels with clinicopathological characteristics was evaluated by using the Wilcoxon rank-sum test. Univariate and multivariate analyses were conducted to determine the prognostic factors, and Cox proportional hazard models were applied to evaluate the patients’ survival outcomes. Spearman’s correlation coefficient and Wilcoxon rank-sum test were performed to calculate the associations of POLD1 expression with the immune cell infiltration scores, marker gene sets of immune status, and immunomodulators. All statistical analyses were conducted using R software (version 4.1.3). Statistical significance was defined as *p* < 0.05 (* *p* < 0.05; ** *p* < 0.01; *** *p* < 0.001).

## 5. Conclusions

Taken together, this study indicates that POLD1 may, as a prognostic biomarker, highlight its novel underlying function in the regulation of immune infiltration and the ferroptosis signaling pathway in ccRCC. The upregulated expression of POLD1 may incur a greater risk of tumor advance in ccRCC patients. In view of the diversity in infiltrating levels and the status of the immune cells between the POLD1^high^ and POLD1^low^ groups, ccRCC patients with POLD1 overexpression may not obtain benefits from more precise immunotherapy strategies in clinics.

## Figures and Tables

**Figure 1 ijms-24-06849-f001:**
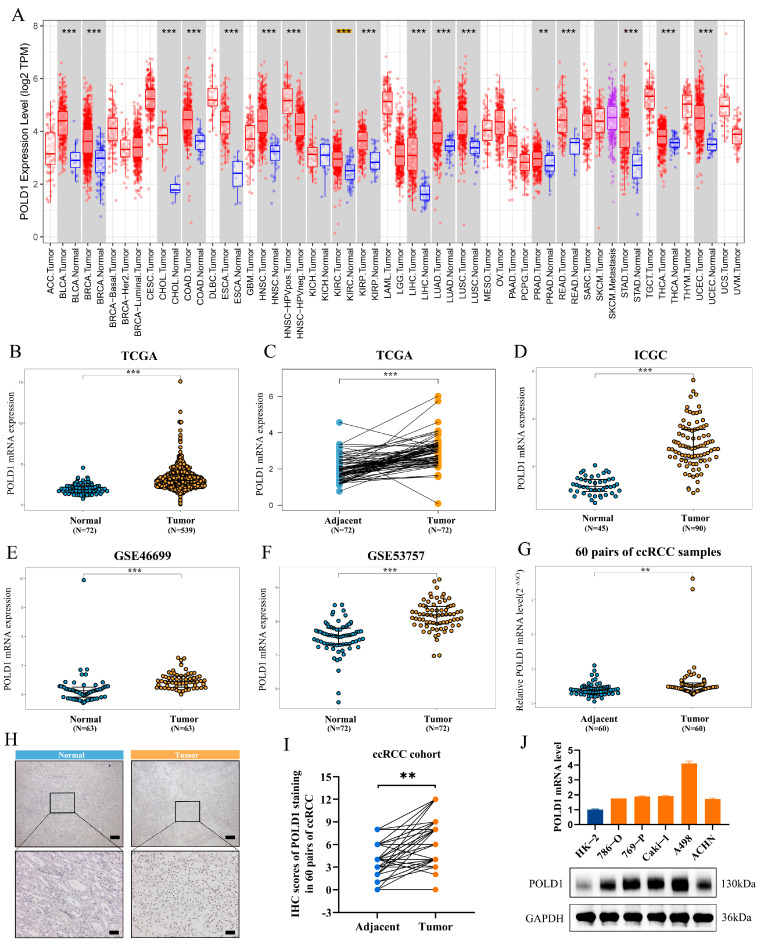
Relationship between POLD1 expression and ccRCC. (**A**) TCGA database analysis showed the POLD1 expression levels in various cancer tissues and their counterpart normal tissues. Yellow background of *** represent KIRC patients. (**B**–**F**) POLD1 expression levels were significantly higher in the ccRCC tissues compared to the counterpart/paired peritumoral normal renal tissues in the (**B**,**C**) TCGA, (**D**) ICGC, (**E**) GSE46699 and (**F**) GSE53757. (**G**) The mRNA level of POLD1 in 60 pairs of ccRCC tissues and their paired normal adjacent tissues. (**H**,**I**) POLD1 IHC staining and statistical results show the protein levels of POLD1 in the validation cohort (scale bar, 50 um; magnification, 100× and 200×). (**J**) The mRNA and protein levels of POLD1 were evaluated in a normal human renal epithelial cell line and RCC cell lines by RT-qPCR and Western blot. Data were shown as mean ± SD. Differences were considered significant at *p* < 0.05 (** *p* < 0.01, *** *p* < 0.001).

**Figure 2 ijms-24-06849-f002:**
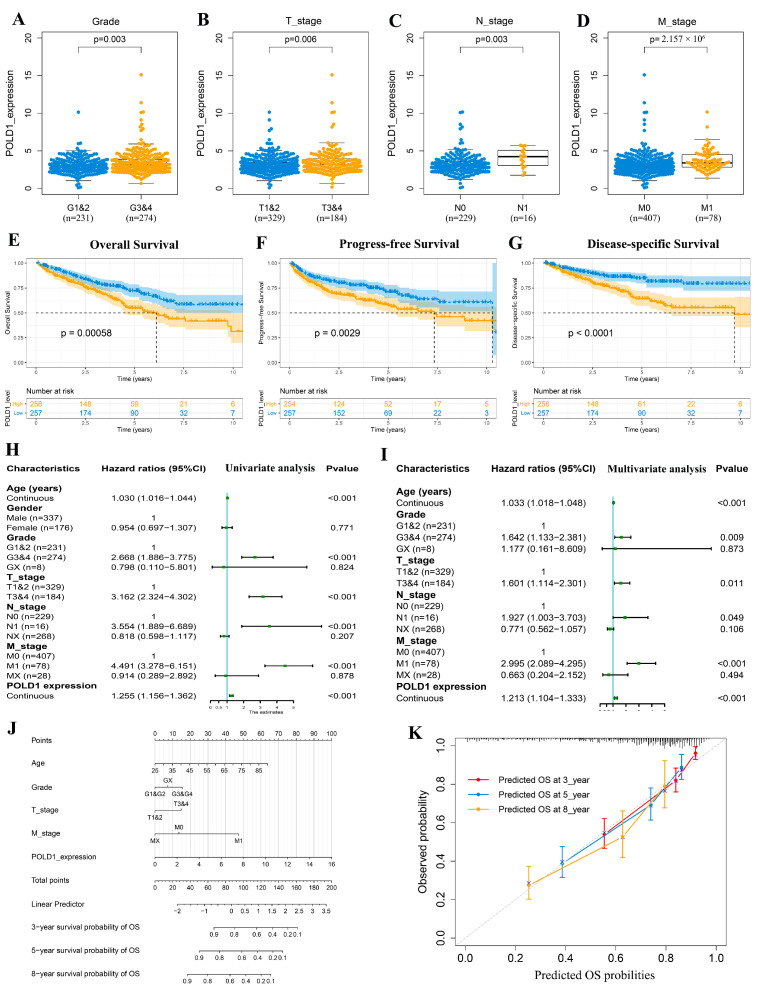
Association between POLD1 expression and the clinical parameters of ccRCC patients and its prognostic significance. (**A**–**D**) Relationship of POLD1 mRNA levels with individual cancer stages (T stage, N stage and M stage) and tumor grade of ccRCC patients. (**E**–**G**) Relationship of POLD1 expression levels with overall survival (OS), progression-free survival (PFS), and disease-specific survival (DSS) in the TCGA-KIRC cohort. (**H**,**I**) Univariate and multivariate survival analyses for selecting prognostic factors. (**J**,**K**) Establishment (**J**) and estimate (**K**) of the overall survival nomogram for ccRCC patients.

**Figure 3 ijms-24-06849-f003:**
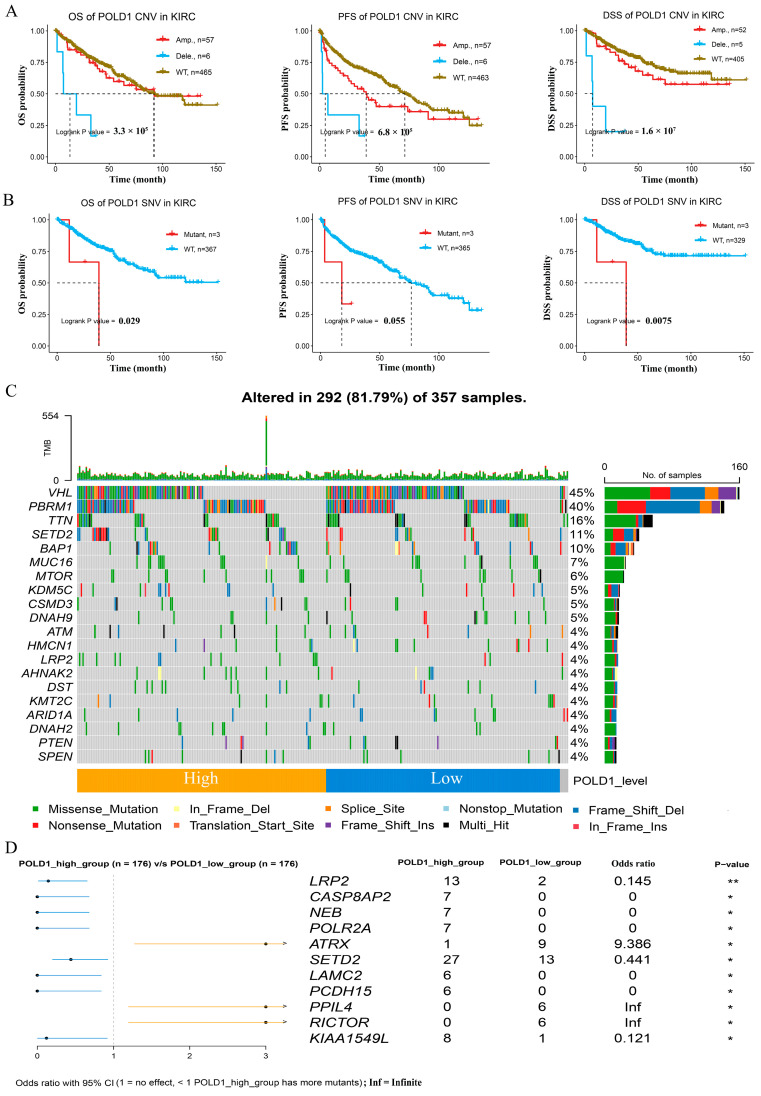
Relationship between somatic mutations and POLD1 expression in ccRCC patients. (**A**) Kaplan–Meier survival curves show the association between the CNV of the POLD1 gene with the OS, PFS, and DSS rates in ccRCC patients. (**B**) Kaplan–Meier survival curves show the association between the SNV of the POLD1 gene and overall survival (OS), progression-free survival (PFS), and disease-specific survival (DSS) in ccRCC patients. (**C**) Somatic mutations in the POLD1-high and POLD1-low expression groups. (**D**) Comparison of the mutations between the high expression and low expression groups of POLD1. * *p* < 0.05, ** *p* < 0.01.

**Figure 4 ijms-24-06849-f004:**
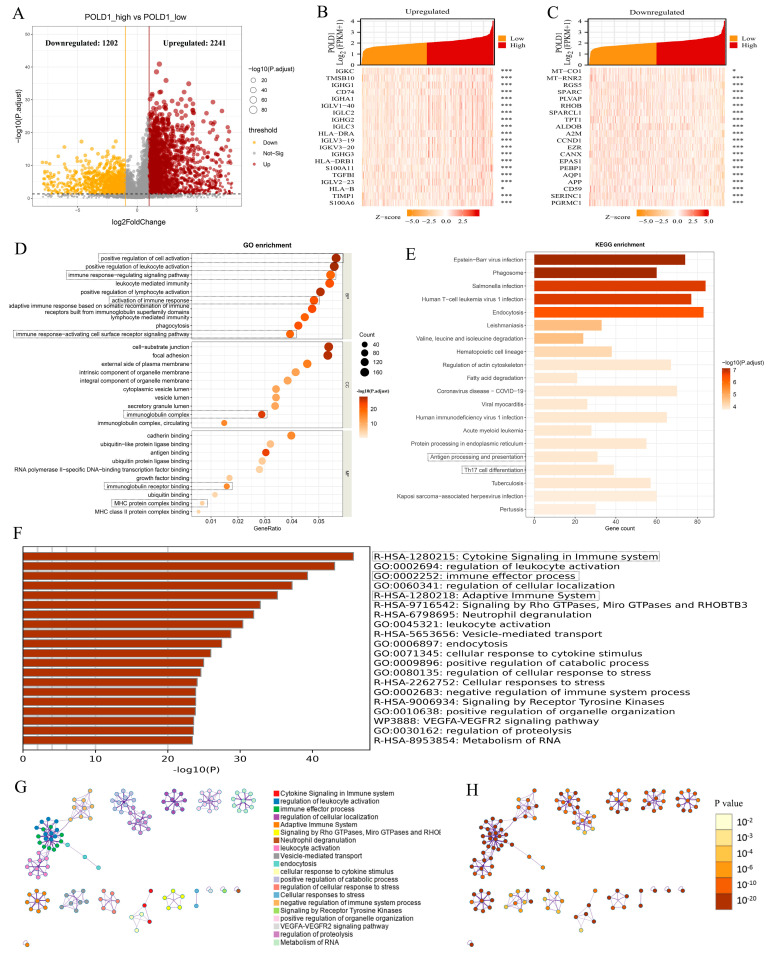
Functional enrichment analysis of the differentially expressed genes (DEGs) based on the POLD1 expression levels in ccRCC. (**A**) The volcano plot described 3443 DEGs (2241 upregulated and 1202 downregulated genes). (**B**,**C**) The heat maps depicted the expression of 20 significant upregulated (**B**) and downregulated (**C**) genes in the ccRCC samples with POLD1^high^ and POLD1^low^ expression. (**D**,**E**) GO (**D**) and KEGG pathway (**E**) enrichment results of DEGs in ccRCC samples with POLD1^high^ and POLD1^low^ expression. (**F**–**H**) The functional enrichment of POLD1-associated DEGs also analyzed via Metascape was shown in (**F**) the bar graph of the top 20 enriched terms, colored by *p*-value. (**G**) colored by cluster ID, where nodes with the same cluster ID are usually close to each other; (**H**) colored by *p*-value, where the terms containing more genes tend to have greater significance. * *p* < 0.05, *** *p* < 0.001.

**Figure 5 ijms-24-06849-f005:**
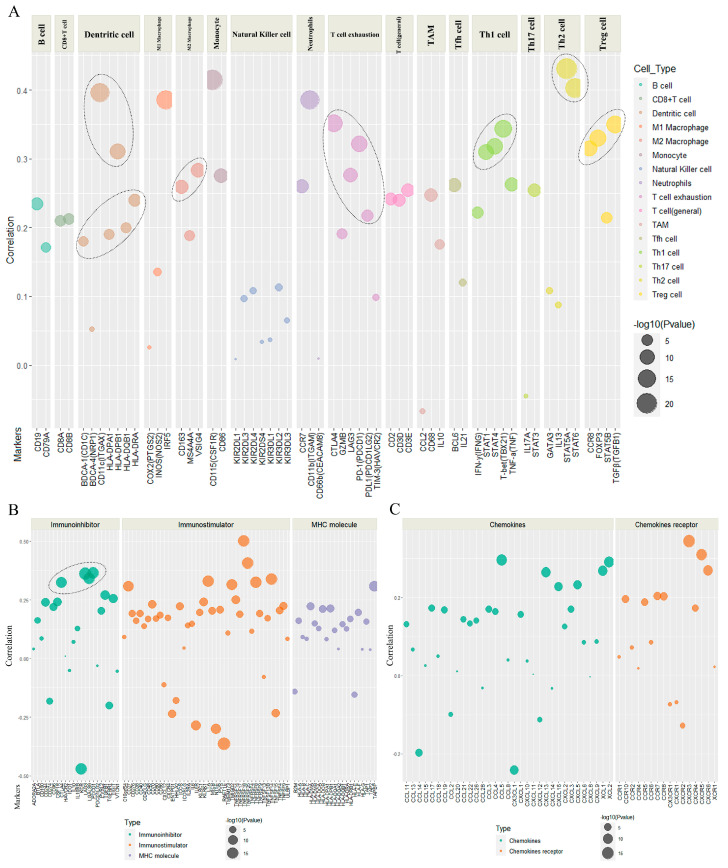
The relationship of POLD1 expression with marker gene sets of immune cells, immunomodulators and chemokines. (**A**–**C**) The bubble plot shows the association of the POLD1 gene with the marker sets of 16 diverse immune cells or their status (**A**); markers of immunomodulators (**B**); as well as markers of chemokines/chemokine receptors (**C**).

**Figure 6 ijms-24-06849-f006:**
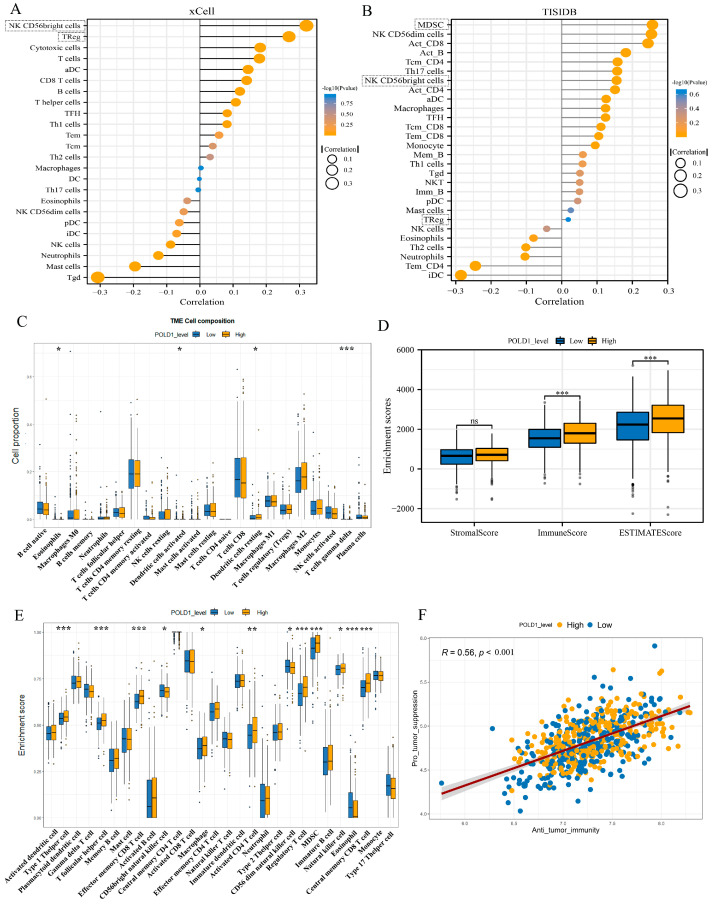
Characteristics of immune cell infiltration in tumors with POLD1 expression. (**A**,**B**) The relationship between POLD1 expression and immune cell infiltrates was analyzed by the xCell (**A**) and TISIDB (**B**) platforms. (**C**) Proportions of 22 immune cells in the subgroups (POLD1^high^ and POLD1^low^) of ccRCC. (**D**) The tumor immune microenvironment (TME) scores between the POLD1^high^ and POLD1^low^ groups (including stromal score, immune score, and estimate score). (**E**) Differences in the activities of diverse immune cells between the POLD1^high^ and POLD1^low^ groups. (**F**) Analysis of the activity of pro-tumor suppression and anti-tumor immunity in the POLD1^high^ and POLD1^low^ groups. ns, No significance, * *p* < 0.05, ** *p* < 0.01, *** *p* < 0.001.

**Figure 7 ijms-24-06849-f007:**
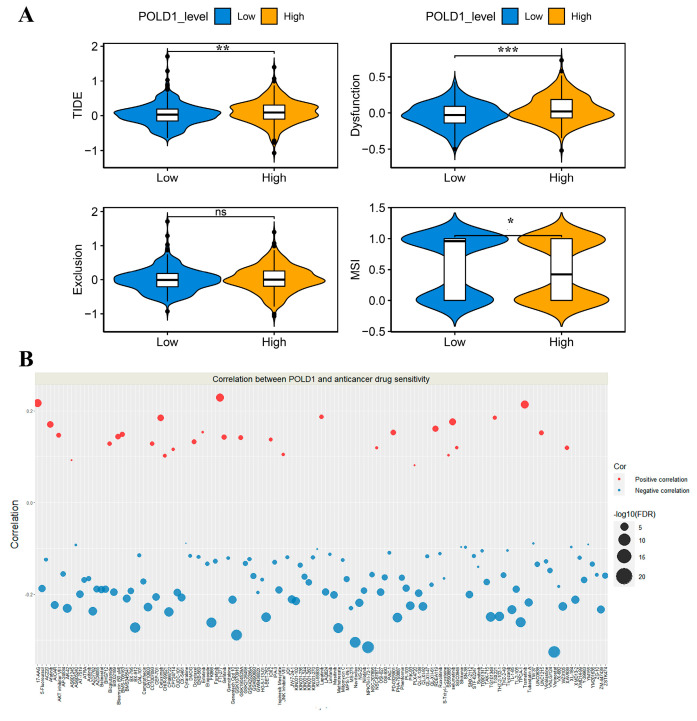
The association between POLD1 expression and anticancer drug sensitivity. (**A**) TIDE score, MSI, T cell Dysfunction and Exclusion in different POLD1 expression groups. (**B**) Gene-set drug-resistance analysis from GDSC and CTRP IC50 drug data. Spearman’s correlation represents an association between gene expression and the drug. A positive association (red bubbles) means that high gene expression confers resistance to the drug, and a negative one (blue bubbles) means it does not. ns, No significance, * *p* < 0.05, ** *p* < 0.01, *** *p* < 0.001.

**Figure 8 ijms-24-06849-f008:**
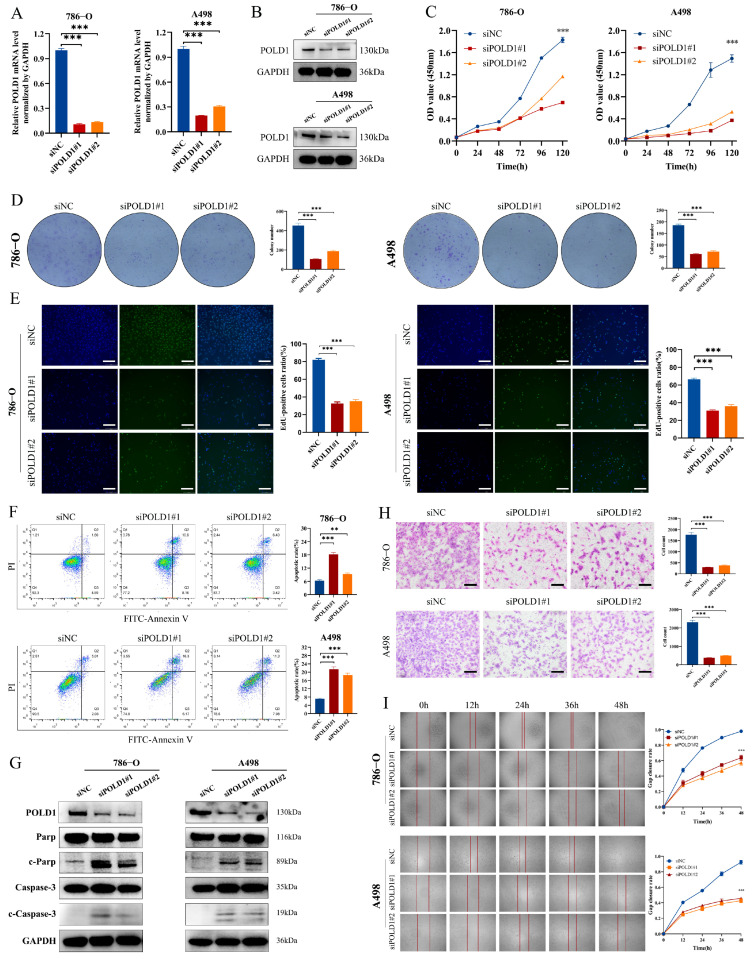
Knockdown of POLD1 inhibited the malignant biological behaviors of ccRCC cells in vitro. (**A**,**B**) The level of POLD1 was evaluated by RT-qPCR (**A**) and Western blot (**B**) after the knockdown of POLD1 in ccRCC cells. (**C**–**E**) The proliferation of ccRCC cells was detected by CCK-8 (**C**), colony-formation (**D**), and EdU assays (**E**). (**F**,**G**) Inhibition of POLD1 resulted in increased apoptosis in ccRCC cells via flow cytometry (**F**) and Western blot assay (**G**). (**H**,**I**) The migration of ccRCC cells was detected by Transwell assay (**H**) and wound healing assay (**I**). Differences were considered significant at *p* < 0.05 (** *p* < 0.01, *** *p* < 0.001).

**Figure 9 ijms-24-06849-f009:**
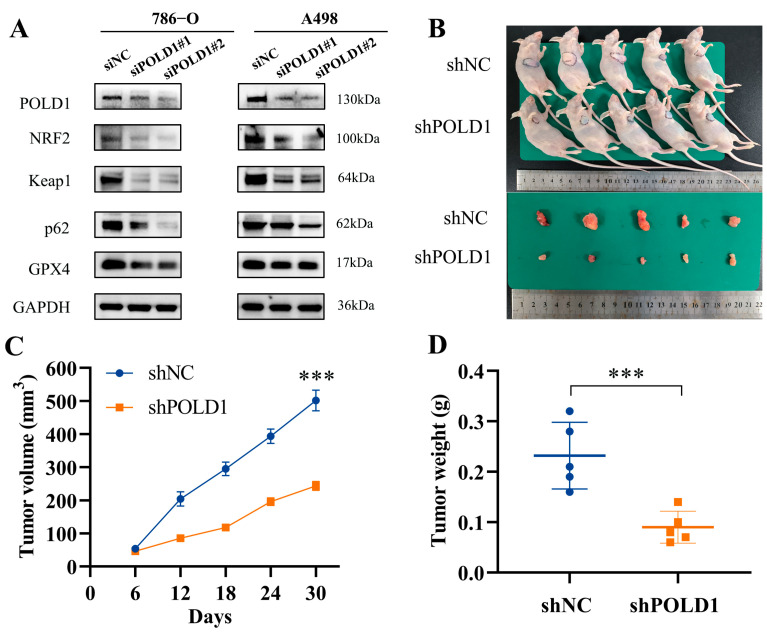
Oncogenic signaling pathways in ccRCC cells and tumor growth in xenograft mouse model after POLD1 knockdown. (**A**) Western blot analysis of Autophagy and Ferroptosis pathway-related proteins with POLD1 knockdown in the 786-O and A498 cells. Tumor growth curve (**C**) of stable POLD1 knockdown 786-O cells (or negative control) in the xenograft mouse model was presented, followed by the collection of tumor nodules (**B**) and tumor weight records (**D**). Differences were considered significant at *p* < 0.05 (*** *p* < 0.001).

## Data Availability

Publicly available datasets and our medical center validation cohort were analyzed in this study. The datasets presented in this study can be found in online repositories. The names of the repository/repositories and accession link(s) can be found in the article/Appendix A. Further inquiries can be directed to the corresponding authors. The informed consent of publicly available datasets was not applicable. The informed consent of our medical center validation cohort was obtained from all subjects involved in current study. Written informed consent has been obtained from the patients to publish this paper.

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
