# Peer review of "POLD1 as a Prognostic Biomarker Correlated with Cell Proliferation and Immune Infiltration in Clear Cell Renal Cell Carcinoma"

_ijms, 2023, doi:10.3390/ijms24076849_

Round 1

Reviewer 1 Report

Comments to the authors: manuscript ijms-2245775

General

The authors studied the role of DNA polymerase delta 1 catalytic subunit (POLD1) as potential biomarker in the cancerogenesis of the clear cell renal cell carcinoma (ccRCC). They evaluated the expression data from various publicly available data bases, among them the TCGA database and validated the summarized results (nomogram) with data collected from an own validation cohort. It is a comprehensive study supported by experimental data (cell culture experiments, animal trials). However, following fundamental concerns regarding the design of this study and differing assessments of the data analyzed require clarification:

Specific concerns

1.    The authors used the TCGA-KIRC expression data from the total ccRCC cohort (and also from the other sources). Thus, both patients with localized and metastatic cancers were included. This undifferentiated consideration of known risk factors (metastasis) seems to be rather unfocused to decipher the possible role of POLD1 in carcinogenesis. It would certainly be useful to focus primarily on localized RCC to limit cohort heterogeneity and prove POLD1 expression as an independent outcome variable with utility compared with traditional clinicopathologic variables. Regional and distant metastases are a priori in this respect unalterable but known "confounders" for the generation of prognostic models in early stage ccRCC. Thus, additional data evaluation should be considered at least.

2.    REMARK guidelines suggest the use of continuous data if they are available. The authors evaluated the POLD1 expression data in association to the clinicopathological variables using continuous data (e.g., Fig. 2A-D) as well as dichotomized data (e.g., high vs. low expression; Supplementary Table 1 and 2). However, contradictory findings resulted, but these results were not discussed and remain confusing for readers. This underlines that a poorly justified dichotomization can lead to serious information losses. It must be clearly pointed out that the univariate/multivariable Cox regression analyses were performed with continuous data. The POLD1 scale in the nomogram indicates that the analyses were obviously performed with continuous data (see expression scale in Fig. 2A-D?). On the other hand, numerous subsequent calculations were performed based on the POLD1-high and POLD1-low expression (e.g., Fig 3-6). Considering the aspect mentioned in comment 1, additional calculations would be necessary.

3.    It is noticeable that the numbers of patients in the various figures did not correspond with the primary patient numbers of the cohorts studied. A general explanation for these differences and repetition in the figures should be given. The patient number (101 subjects) in Supplementary Figure 1 is also incomhensible, although the validation cohort has only 60 patients.

4.    Reference 24: Godlewski et al. also examined the POLD1 expression in ccRCC but found increased POLD1 expression as favourable marker for overall survival. This point should be discussed as this result contrasts with the presented data here.

5.    Minor point: the word "correlation" should only be used if correlation coefficients are actually calculated, otherwise "association" should be preferred.

Reviewer 2 Report

the paper covers an interesting issue regarding biomarkers in mRCC. There are some revisions and limits that should be explained and improved.

- Authors state that POLD1 maybe a prognostic biomarker for OS and PFS but no data are reported regarding the treatment received by patients. This could be a bias and is a major limit of the study. Therefore, I suggest the paper by Simonetti et al (PMCID: PMC9476128) that reports as RANKL is a biomarker of reponse in mRCC patients treated with nivolumab.

- In the introduction it would be interesting to cite more data regarding mRCC biomarker and the ongoing progress regarding molecular clustering of RCC ( Motzer et al Cancer Cell 2020 ; Denize T et al PMID: 35802667)

Round 2

Reviewer 1 Report

All critical comments have been convincingly addressed in the revised version; no further objections.